# Modeling to Understand Plant Protein Structure-Function Relationships—Implications for Seed Storage Proteins

**DOI:** 10.3390/molecules25040873

**Published:** 2020-02-17

**Authors:** Faiza Rasheed, Joel Markgren, Mikael Hedenqvist, Eva Johansson

**Affiliations:** 1Department of Plant Breeding, The Swedish University of Agricultural Sciences, Box 101, SE-230 53 Alnarp, Sweden; Faiza.rasheed@slu.se (F.R.); Joel.markgren@slu.se (J.M.); 2School of Chemical Science and Engineering, Fibre and Polymer Technology, KTH Royal Institute of Technology, SE–100 44 Stockholm, Sweden; mikaelhe@kth.se

**Keywords:** albumin, globulin, glutelin, monte carlo simulation, molecular dynamics simulation, prolamin

## Abstract

Proteins are among the most important molecules on Earth. Their structure and aggregation behavior are key to their functionality in living organisms and in protein-rich products. Innovations, such as increased computer size and power, together with novel simulation tools have improved our understanding of protein structure-function relationships. This review focuses on various proteins present in plants and modeling tools that can be applied to better understand protein structures and their relationship to functionality, with particular emphasis on plant storage proteins. Modeling of plant proteins is increasing, but less than 9% of deposits in the Research Collaboratory for Structural Bioinformatics Protein Data Bank come from plant proteins. Although, similar tools are applied as in other proteins, modeling of plant proteins is lagging behind and innovative methods are rarely used. Molecular dynamics and molecular docking are commonly used to evaluate differences in forms or mutants, and the impact on functionality. Modeling tools have also been used to describe the photosynthetic machinery and its electron transfer reactions. Storage proteins, especially in large and intrinsically disordered prolamins and glutelins, have been significantly less well-described using modeling. These proteins aggregate during processing and form large polymers that correlate with functionality. The resulting structure-function relationships are important for processed storage proteins, so modeling and simulation studies, using up-to-date models, algorithms, and computer tools are essential for obtaining a better understanding of these relationships.

## 1. Introduction—Plant Proteins: Types, Characteristics, and Presence

### 1.1. The Variety of Plant Proteins and their Functions

Proteins are biological macromolecules, that are responsible for a majority of the biological roles (sometimes together with other biomolecules) in the cell. Thus, proteins can be considered biomolecular devices with natural structural and functional properties that are often challenging to recreate in the laboratory. Some classical examples of proteins with exceptional functions are spider silk proteins, combining very high strength with excellent elasticity [1], and bacterial flagella, which are microscopic propellers [2]. Proteins in the form of enzymes catalyze reactions at rapid rates with great selectivity.

Like proteins in general, plant proteins play various enzymatic, structural and functional roles (photosynthesis, biosynthesis, transport, immunity, etc). They also act as storage mediums to meet the growth and nutritional demands of developing seedlings. Proteins perform these functions in their composition and specific structural forms, e.g., through folding, which can range from compact and well-ordered to unfolded and intrinsically disordered.

### 1.2. Classifications of Plant Proteins—Relationships to Seed Storage Proteins

The first attempts to classify plant proteins were based on the extractability and solubility of these proteins, with the first systematic study performed on seed storage proteins by T.B. Osborne, applying the classification scheme suggested by the American Committee on Protein Nomenclature [3]. This nomenclature basically classifies proteins into three types; simple, conjugated, and derived. Proteins in all plant tissues are classified as simple proteins, which are further divided into four types [4] (Table 1). These four types of plant proteins are mainly associated with seed storage proteins and are known as albumins, globulins, prolamins and glutelins, separated based on Osborne fractionation with water, salt, alcohol, and alkali, respectively (Table 1). Later attempts at more complex classifications of plant proteins have been made, based, e.g., on their chemical structure, mechanism/s of actions, biological function or within-plant location. Despite these attempts to more modern classification systems, Osborne classification is still the most widely used system, particularly used for protein extraction and purification procedures [5]. In practice, however, the Osborne classification has only been used on seed storage proteins, while classification of other plant proteins is generally more complex and sometimes unclear.

### 1.3. Characterization and Presence of Osborne Classified Proteins

Albumins are defined as water-soluble, globular proteins that are coagulable by heat (Table 1). The most well-known albumins are serum albumin, the major protein in human blood, and egg white. In plants, albumin is present as a 2S albumin storage protein in seeds, e.g., as leucine in barley, wheat, and rye, as legumelin in pea, soybean and cowpea, as phaselin in kidney bean, and as ricin in castor bean [6] (Table 1). Many proteins in the green plant tissues, including ribulose-1,5-bisphosphate carboxylase oxygenase (RuBisCO), an enzyme catalyzing the first step of carbon fixation and the most abundant protein on Earth, are not defined as albumins, despite being water-soluble. RuBisCO is water-soluble and coagulable by heat [12], which should by the Osborne definition, make it an albumin protein. Similarly, the majority of the enzymatic proteins in plants are water-soluble and coagulable by heat, but apart from being characterized as enzymes, they have not been further defined into a protein type.

Globulins are also globular proteins, which have a higher molecular weight than albumins, and are soluble in dilute salt solution, but insoluble in water (Table 1). The most well-known globulin is a major human blood protein (serum gamma globulin) [13]. In plants, globulins are present as storage proteins in both, dicots and monocots, making them the most common group of storage proteins [7]. Based on sedimentation coefficient, the plant storage globulins are basically divided into two groups, 7S vicilin-type, which has been found and extensively studied in pea, soy bean etc., and 11S legumin-type, which is, e.g., the major storage protein in most legumes and in dicots such as brassica, oat, and rice [7] (Table 1). Some plant leaf proteins are clearly not directly water-soluble and may be classified as globulins, in some characterizations of leaf proteins [14]. However, the reason that these leaf proteins are not water-soluble might be because they are cell wall-bound, interact with pectin, or are hydrophobic [15], and thereby, obviously not easy to categorize as globulins or albumins.

The additional two protein types, prolamins and glutelins, are found in particular as storage proteins in seeds of the grass family (Triticeae), where they are the dominant proteins, comprising up to 85% of total protein [7] (Table 1). The prolamins found in wheat are called gliadins, while the nomenclature of the prolamins in other cereals is based on their Latin names; zein in maize, hordein in barley, secalin in rye etc. [8] (Table 1). The most commonly found glutelin is that found in wheat (glutenin), although glutelins are also present in barley and rye [11] (Table 1). Prolamins and glutelins have several similarities, including a high proportion of proline and glutamine, and a high proportion of repetitive motifs or sequences, with non-repetitive domains in their N- and C-terminals [8]. Although, these proteins differ in molar mass, the major difference between the two protein types is the formation of intra- and inter-molecular disulfide bonds in the native state of prolamins and glutelins, respectively [9], which explains the differences in their extractability.

### 1.4. Seed Storage Proteins—Types and Characteristics

From the previous sections, it is clear that seed storage proteins are present in the seed in various types, including albumin, globulin, prolamin and glutelin [6,7,8]. These proteins may have different characteristics in the plant cell and also during processing and for various applications, where the storage proteins contribute functionality. However, to date, the seed storage proteins have mainly been characterized based on their chemical performance during fractionation, sedimentation, etc. [4,6,7,8]

## 2. Functionality of Plant Proteins—In the Plant and for Food and Industrial Applications

### 2.1. Function of Plant Proteins in the Plant

Plant proteins meet the needs of the emerging seedling in terms of nutrition and growth, through their enzymatic, structural, functional and storage functions [16]. Plants contain a number of specific types of proteins not found in other living organisms, and these have certain functions (Table 2). For example, most plants have some kind of storage organ (seeds, tubers etc.) for reproduction, where different nutrient sources are stored so that the new plant will have the resources to grow during the coming season. Proteins, carbohydrates, and oils are different types of nutrient sources accumulated in plant storage organs. Such proteins are normally described as storage proteins (Table 2). Their primary function is to be broken down into amino acids, to form the necessary building blocks for emerging proteins in the next-generation growing plant [17]. The plant cell contains a number of organelles, including chloroplast, which are responsible for photosynthesis (Table 2). Plants also have a specific protein, the enzyme RuBisCO (Table 2), for catalyzing the transfer of solar energy to chemical energy that can be used by the plant, through CO_2_ fixation [18].

### 2.2. Functions of Plant Proteins in Food

In food products, proteins are often the main functional component, especially in processed food products with a high protein content [19]. In such products, proteins contribute to nutrition, but also to food quality, texture, aroma, flavor, feeling of satiety, and ease of processing [20] (Table 2). Most plant proteins have characteristics that make them interesting in food processing although their specific characteristics may differ. For example, the pea protein vicilin has been shown to have better heat-induced gelation and emulsifying properties than the pea legumins, which instead have been found to be more nutritious for humans [21]. The performance of various plant proteins for food applications is highly dependent on their structural features and their ability to form specific three-dimensional (3D) configurations/conformations, as well as their ability to cross-link [10,22].

### 2.3. Functions of Plant Proteins in Industrial Applications

Plant proteins also have properties that make them interesting for use as materials (Table 2). A range of plant proteins, including mainly storage proteins from e.g., wheat, soybean, potato and oilseed crops, have interesting properties for applications such as packaging, fire resistant and absorbent materials [23,24,25,26,27,28,29,30] (Table 2). The ability of proteins to cross-link and aggregate is important for good protein-based material properties [10,31,32,33].

### 2.4. Impact of Seed Storage Proteins on Functionality

The descriptions above clearly show the primary importance of the storage proteins in plants as a source of amino acids to be used as building blocks for the developing young seedling at emergence [17]. In food and industrial applications, seed storage proteins play a central role for functionality [19,20,21,22,23,24,25,26,27,28,29,30], making it important to understand how the functionality is influenced and can be fine-tuned [35,36,37].

## 3. Modeling—State of the Art for Plant Proteins

### 3.1. Basis for Modeling to Study Protein Structures

As described above, the three-dimensional (3D)-conformation of native proteins, together with their ability to crosslink, reshape, and form specific structural features, contribute substantially to the functional properties of specific proteins in the plant and in plant-products [31,34,38]. The methods for determining the structure of proteins can essentially be divided into three types, i.e., those based on different types of microscopy techniques, those based on scattering/diffraction and spectroscopy (e.g., X-ray and nuclear magnetic resonance (NMR) techniques), and those based on modeling [39] (Table 3). In this review, we focus on the opportunities and drawbacks of using modeling to determine plant protein structures, with specific emphasis on plant storage proteins.

### 3.2. Template-Based and Ab Initio Modeling

The methods applied to model plant protein structures are similar to those used for any type of protein (Table 3). Template-based modeling is the most simplified method for protein modeling. Template-based models are built on comparisons of amino acid sequences from proteins with known protein structures (often identified by crystallography) and the assumption that similar sequences result in the same protein structures [40]. However, many proteins do not share sequence similarities with other proteins already present in data-bases and with known structures. For such proteins (showing no relationship with known proteins), modeling from sequence information alone, i.e., ab initio protein structure prediction, is the only option. In ab initio predictions, thermodynamic principles are applied to the case of protein conformation, through a search for the overall energy minimum [40].

### 3.3. Monte Carlo, Molecular Dynamics and Machine-Learning Methods

Two more advanced/modern methods, Monte Carlo (MC) and Molecular Dynamics (MD) have been developed during recent years for protein structure predictions and simulations. Both methods resemble the ab initio approach as modeling is based on sequence information alone. (Table 3). Both methods can be applied on different scales with different degrees of detail; from all-atom and united atom to coarse-grain models [41]. The advantage with MC computations is that it is faster and easier to perform than MD computation, due to the fact that it is free from the restriction of solving Newton’s equation of motion [42] (Table 3). However, this also affects the results, since no “dynamic” information is gathered from the MC run [42,43]. Thus, in several protein simulations, MC has been combined with MD [44]. Furthermore, modern machine-learning techniques, such as the algorithm AlphaFold, have brought great advances in prediction of protein structure from sequence information [45]. In the most recent Critical Assessment of Protein Structure Prediction (CASP) experiments, the AlphaFold 3D models of proteins were placed first in the Free Modeling category, in terms of accuracy [46]. The AlphaFold algorithm uses artificial neural networks to build a protein-specific fragment library [47]. However, algorithms, such as AlphaFold have not yet been applied to predict the structure of seed storage proteins, probably because these algorithms generally predict protein structures, based on folding, while some of the most useful storage proteins are intrinsically disordered.

### 3.4. Modeling Plant Proteins with Specific Emphasis on Seed Storage Proteins

The plant protein modeling research field is comparatively small and relatively new, with a limited number of plant-related deposits in the Research Collaboratory for Structural Bioinformatics Protein Data Bank (RCSB PDB). A search for plant deposits currently results in 2300 hits, with one quarter from Arabidopsis. Sorting, based on the most common organisms, shows 1382 hits on Arabidopsis and 15000 hits on others (others = plant based organisms but also additional organisms) in an overall total of more than 160 000 total deposits (the majority coming from humans and microorganisms) [57]. One third of the deposits in the current plant search are from 2015 and later. Only 47 deposits are available for seed storage proteins, the majority being globulins and the rest albumins. These deposits indicate that plant protein modeling has focused on specific areas. This is also reflected in by the information in the RCSB PDB, where a high number of deposits are available for example, in proteins associated with the photosynthetic machinery, proteins associated with ribosomes, enzymes, stress and defense, and for allergens and sweet-tasting proteins [57]. Plant protein modeling, within these different areas, is summarized in Section 4.1, Section 4.2, Section 4.3, Section 4.4, Section 4.5 and Section 4.6, in order to identify opportunities of relevance for modeling seed storage proteins.

## 4. Main Plant Protein Modeling Areas and Impact for Modeling of Seed Storage Proteins

Modeling approaches for six of the most common plant proteins in the RCSB PDF are described below. Seed storage proteins are not among these, although knowledge derived from other areas of research might be of relevance for modeling seed storage proteins.

### 4.1. Photosynthetic Machinery

The photosynthetic machinery is responsible for the transfer of solar energy to chemical energy through CO_2_ fixation [18]. It is, thus, one of the most important and specific traits of plants, and has long attracted the interest of the scientific community. MD simulations, complemented with quantum mechanical descriptions, were applied already in the early 1990s to describe how electron transfer is controlled by protein motion in photosynthetic reaction centers [58]. Molecular dynamics simulations have since contributed to the understanding that protein movement in the reaction centers is key to the kinetics of the primary electron transfer reaction [59]. To understand the full mechanisms of the photosynthetic machinery, a range of both, analytical and simulation methods have been adopted, as recently reviewed by Blumberger [60,61]. The tools for assessing protein movements in reaction centers might be useful for investigating seed storage protein movements during processing and dynamics/energy transfers for these movements. The structure of RuBisCO, and of divergent, mutant, and hybrid forms of this enzyme, has been characterized through X-ray crystallography [62]. Molecular dynamics simulations have also been used to explain variations in the functionality of RuBisCO mutants, where the use of structural checkpoints has been found to enable fine-tuning of the dynamics of the enzyme [63]. Similarly, dynamics of seed storage proteins already characterized by experimental methods such as X-ray crystallography can be understood by computer-based simulation methods.

### 4.2. Proteins Associated with Ribosomes

Cryo-EM studies, combined with X-ray crystallography, have been applied to determine the structures of ribosomes, producing static models for the various states of ribosomes. Recently, computational studies involving simulations have been successful in shedding light on structural fluctuations and transitions among the different ribosomal configurations [64]. Advances in MD simulations, including large-scale MD, are one explanation for the successful simulation of large macromolecular complexes, such as ribosomes [65]. The size of some seed storage proteins creates challenges in their modeling, resulting in similarities with the modeling of ribosomes. Thus, the use of large-scale MD might be an alternative for modeling seed storage proteins.

### 4.3. Enzymes

Mathematical modeling approaches have long been a useful tool for investigating the complexity of metabolic networks and their enzymatic regulation, while more recent models have contributed greatly to the growing field of systems biology [66]. Most commonly, enzyme-kinetic models have been applied to examine enzymatic regulations [66], although more recently, MD techniques and molecular docking simulations have been used for similar purposes [67]. The enzyme-kinetic modeling of metabolic pathways differs substantially from the modeling of structures and functions in seed storage proteins.

### 4.4. Stress and Defense

Regarding stress proteins and defense mechanisms, simulations have been used to gain a better understanding of the mechanisms involved [68,69,70,71]. Again, recent uses of MD simulations has improved our understanding of these proteins. Both homology modeling and MD simulations have been used to examine the background to herbicide resistance in plants [72]. Homology modeling, molecular docking, and MD simulations have been used to assess differences in protein conformations contributing to resistance compared with susceptible reactions in plants to different diseases [73]. The development of the highest relevance for modeling seed storage proteins is methodology describing differences in protein conformation related to different functions and functionality.

### 4.5. Allergens

Allergens are often proteins [74] and plants contain a variety of allergens towards which sensitive humans display allergic reactions [75]. Plant proteins responsible for allergic reactions have been structurally modeled using crystallography, X-ray scattering and NMR, as well as docking simulations of protein models with similar sequence [76,77]. Further, the 3D crystal structure of various plant-derived allergy proteins has been determined and MD simulations have been used to detect molecular conformations of the proteins involved in allergic reactions [78,79,80]. Several plant-based allergens are also seed storage proteins in plants, so modeling carried out on allergens might be directly transferable to research on seed storage proteins and structure-function relationships.

### 4.6. Sweet-Tasting Proteins

Sweet-tasting proteins are specific plant-based proteins of great interest as they have characteristics making them hundreds to thousand times sweeter in taste than sugar [81,82]. Several studies have focused on investigating this property, using techniques, such as crystallization of the proteins and determination of their structure with NMR and X-ray crystallography [81,83]. Comparative/homology modeling and molecular docking techniques have been used to predict 3D structures of dimer and tetramer forms of some sweet proteins, while the effects of pH on protein conformation have been evaluated using MD simulations [82]. MD simulations have also been used as a tool in structure-guided protein engineering for designing improved low-calorie plant-based sweeteners for pharmaceutical and food applications [84]. The methods applied to understand structure-function behavior and those used for structure-guided engineering might be of interest for seed storage protein research and applications.

## 5. Understanding Structure-Function Relationships of Seed Storage Proteins

Plant storage proteins are probably the second most abundant protein group in plants (after RuBisCO). In all plants containing storage organs, such as grains and seeds, the function of the storage proteins is to store amino acids necessary for growth and development of the emerging seedling [85]. Few investigations with modeling tools have been performed on these proteins, despite the abundance of plant storage proteins, their importance as a source of nutrition for the emerging young plantlet [85], and their impact on the functionality of products from plant grains [10,22,23,86,87,88,89]. However, bioinformatics techniques are increasingly being employed for classification of different plant proteins, with neural networks displaying accuracy of 95.3% in classifying rice proteins into different classes (albumins, globulins, prolamins, glutelins) [90]. Additionally, machine-learning algorithms have been successfully used to classify seed storage proteins from rice, wheat, maize, castor bean, and thale cress into their classes [91]. Structurally, the different classes of seed storage proteins can be divided into two types, albumins/globulins and prolamins/glutelins (Figure 1). Albumins and globulins are generally highly structured and thereby able to crystallize, and their folding can be simulated using a range of methods, including machine learning and ab initio modeling [92,93,94,95,96,97,98] (Table 4). Most prolamins and glutelins are instead intrinsically disordered [99], and thereby, pose more challenges in modeling. They would require MC- and MD-based algorithms for modeling their structures, although small-angle scattering methods combined with infrared spectroscopy (IR) and high-performance liquid chromatography (HPLC) have also been applied to examine structural changes during processing [100,101,102,103,104].

### 5.1. Albumins

Crystal structures have been identified for a few albumin storage proteins using NMR and X-ray crystallography [92], and two of these protein structures have been deposited in the RCSB PDB [57]. Limited information is available on structural modeling using simulation tools for grain storage albumin proteins.

### 5.2. Globulins

Practically all deposits in the RCSB PDB on plant globulins are based on X-ray diffraction. The fact that X-ray diffraction-based models are available, enables simulation-based verification of the protein structures and simplifies further computer-based modeling. Globulins from legume seeds were the first storage protein to be crystallized and evaluated with X-ray diffraction [93,94,95,96]. The crystal structure was found to be compact with salt bridges and hydrophobic clusters, resulting in layers of packed molecules forming aggregates [94]. Since then, homology modeling has been used to model other globulin proteins, e.g., vicilin in cocoa, based on crystal structures of legume globulins such as jack bean canavalin and French bean phaseolin [97] (Table 4). These studies indicate that hydrophobic amino acids are buried inside the protein molecule at trimer formation, while histidine residues are found at the interfaces towards other globulins [97]. Later studies used homology modeling of *Arabidopsis thaliana* to identify the structure of the oilseed storage protein cruciferin [98] (Table 4). The impact of structure on the function of cruciferin has been evaluated through the use of different isoforms of the protein [98]. Recent studies using 3D molecular models and computational simulations have demonstrated ability of vicilin-like proteins from leguminous plants to bind to chitin or chitinous structures through three chitin-binding sites at each vicilin trimer [106]. Examples of structural features of the storage protein 7S globulin in soybean and 11S globulin in pea, obtained through simulation based on amino acid sequences [39,107,108] of single subunits and their polymers, are shown in Figure 1.

### 5.3. Prolamins

Prolamins and glutelins have been less studied by modeling and simulation than the globulins and most other types of proteins. The major reason for the lack of models on prolamin proteins is that their structure is intrinsically disordered [99]. Solubility studies, using various solvents, have shown that the prolamins are monomeric in their natural stage [4,10] (Figure 1). However, during processing, the prolamins have been shown to form disulfide bonds with other seed storage proteins, thereby, contributing to the formation of polymers [10,109]. Early studies evaluating the structural features of the prolamins in wheat (the gliadins), using Fourier transform infrared spectroscopy (FT-IR), detected equal amounts of α-helix, linear β-structure, β-turns, and unordered structure [110]. The structures of the gliadins have been evaluated at pH 3.0 with dynamic light scattering, cryo-transmission electron microscopy, and small-angle X-ray scattering, followed by ab initio prediction and MD simulations [105] (Table 4). These studies indicate the presence of dimers with a hydrodynamic radius of 5.72 nm, aggregated clusters of 30 nm, and oligomers of 68 and 103 nm [105]. Importance of the concentration of gliadins in distilled water for their state of aggregation has also been reported [103]. At 0.5 wt-% gliadin, repulsion of the gliadin assemblies can be observed, resulting in the protein mainly being present in its monomeric form, with only limited amounts of dimers and oligomers, while at 15 wt-% gliadin, a gel-like hydrated solid is formed as a result of formation of aggregates [103]. Under heat and pressure, the gliadins have also been shown to form hexagonal structures with a 65 Å lattice parameter [27,33,38]. Ab initio modeling has been used to investigate the reasons for formation of hexagonal structures by the gliadins, and has partly ascribed it to glycerol acting as a chemical chaperon aiding in the packing of the protein molecules [38]. The prolamins in maize (zeins) have been modeled with MD simulations to a greater extent than the wheat gliadins. A structural feature reported for the zeins is an α-helix with four amino acid residues per turn and a hydrophobic face inside the helix, formed by the non-polar residues with the carotenoid lutein, helping to stabilize the structure [111]. During solvent evaporation, the zeins are able to self-assemble into different protein shapes, including rods, spheres, and films of different sizes, partly due to so-called head-to-tail binding of the proteins [112]. Early studies on wheat suggested head-to-tail binding [113,114] of the gliadins while aggregating, which has also been shown in confirmed studies [103,105].

### 5.4. Glutelins

The glutelins are even larger and more complicated molecules than the prolamins [104] (Table 4), and wheat glutelins (glutenins) are known to form the largest polymers in nature [115] (Figure 1). This is one reason for the limited modeling of these proteins. The vast majority of previous studies have focused on examining how these large polymers are formed and the background to their formation [10,31]. Depolymerization and re-polymerization of the gluten polymer are known to occur during processing operations, such as dough mixing [116]. Therefore, theoretical models of polymerization mechanisms have been one way forward in using modeling tools to predict the glutenin polymer structure [117]. Through these models, the directionality of formation of the polymers (head-to-tail, head-to-head, tail-to-tail or tail-to-head) can be evaluated. In a recent study using HPLC and small-angle X-ray scattering, a highly intrinsic and disordered structure in the native glutenin protein of wheat was reported, while the structures of unknown types were formed during processing of the proteins into films [34]. Molecular modeling, homology modeling and MD simulations have also been used on the *N*-terminal domain of glutenins to examine the polymerization of the proteins into giant oligomers through disulfide bond formation [118,119].

### 5.5. Future Opportunities for Modeling Cereal Seed Storage Proteins

The main reasons for the lack of studies, using modeling tools and MC/MD simulations to investigate the prolamin and glutelin proteins, are that the (i) proteins are large and among the largest proteins in nature [120]; (ii) proteins in their native form are intrinsically disordered and not soluble in water [99]; (iii) structures of the proteins change with dilution [99] or processing [31], i.e., into low levels of structures in dilute condition [103], hierarchical structures in concentrated regimes [99], and hexagonal [27,31] and other types of structures [34] developing under certain processing conditions; and iv) reported structures are always connected to aggregation of the proteins into polymers [10]. Thus, modeling these proteins is extremely difficult and requires a large amount of computer power. However, the storage proteins of cereals are highly useful and important: (i) as a storage resource for the plant to regenerate itself [7]; and (ii) for human [7] and animal [121] food and feed purposes; and are potentially useful in (iii) non-food applications, such as replacement of petroleum-based plastics [10,122]. A clear structure-function relationship has been demonstrated between the formation of protein structures during processing and the functionality of products produced from cereal storage proteins [123,124]. Therefore, a better understanding of the structural features of the plant storage proteins, especially for the under-researched prolamins and glutelins, and opportunities to fine-tune their structural performance would contribute significantly to their utility in various applications. Furthermore, such an understanding helps reveal the physiological and evolutionary reasons for variations in storage proteins among plants. Novel modeling tools, together with faster/more powerful computers and computer cluster tools to empower the simulation algorithms, are of utmost importance in such work, and will open up opportunities for identifying even the largest and most complex protein structures. Novel simulation tools and newly-developed algorithms will be available for researchers and others in the near future, together with stronger computers and computer clusters, which will pave the way for modeling, even the largest and most intrinsically disordered proteins, such as seed storage glutelin proteins. In a future perspective, research on gliadins and glutenins might also benefit from a similar approach to the approaches used for elastomeric proteins, utilizing low complexity (not coarse-graine) models and/or work with peptide fragments to reveal protein behavior [125]. Such modeling will create novel opportunities for assessing biological features in plants and for fine-tuning the properties of foods and materials produced from these proteins.

## 6. Conclusions

A combination of innovations within computing technology, increased speed, and power of computers, and novel modeling/simulation tools will increase the opportunities that determine protein structures and reveal structure-function relationships. Until recently, modeling and simulation tools have been rarely used for the evaluation of plant protein structures and structure-function relationships. Thus, less than 9% of the current deposits in the Research Collaboratory for Structural Bioinformatics Protein Data Bank are plant-related and one-third of these deposits are less than five years old. For the two most abundant plant protein types, RuBisCO and the storage proteins, modeling has been used to different extents. For RuBisCO and the photosynthetic machinery, modeling has been used rather frequently to describe the process and structural impacts of changes in the protein. Structural features of the storage proteins have been described to a lesser extent by modeling, especially for the prolamin and glutelin. The present status of modeling as regards structural features of the storage proteins is exemplified in Figure 1 for globulin (soybean, pea), prolamin (wheat α-gliadin), and glutelin (wheat glutenin) proteins, respectively.

## Figures and Tables

**Figure 1 molecules-25-00873-f001:**
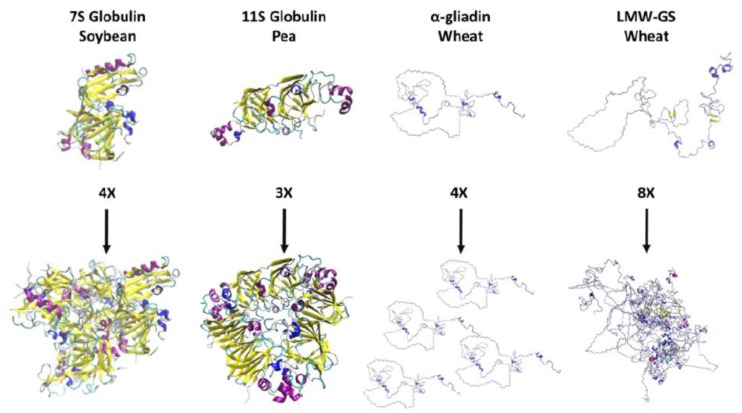
Single subunits and polymers obtained from these, of (left to right): The storage proteins 7S globulin in soybean, 11S globulin in pea, a-gliadin in wheat, and low molecular weight glutenin subunits (LMW-GS) in wheat. Amino acid sequences and simulation tools applied to determine the structures are described in Yoshizawa et al. [107] and Research Collaboratory for Structural Bioinformatics Protein Data Bank (RCSB PDB) (3AUP) [39] for 7S globulin, and in Tandang-Silvas et al. [108] and RSCB PDB (3KSC) [39] for 11S globulin. Amino acids sequences for a-gliadin and LMW-GS can be found at Uniprot accession numbers Q9ZP09 and P10386 and simulation was carried out using an all-atom model with Monte Carlo algorithms in the program Profasi [126]. All models were visualized using Visual Molecular Dynamics [127].

**Table 1 molecules-25-00873-t001:** Types, characteristics (based on Osborne fractionation [3,4]), and presence of plant proteins.

Protein Type	Solubility	Characteristics	Examples in Plants	References (examples)
Albumins	Water	Globular, coagulable by heat	2S-type; e.g., Leucine, Legumalin, Phaselin, Ricin	[4,5,6]
Globulins	Salt solution	Globular, higher molecular weight than albumins	7S vicilin-type (peas, soy etc.) 11S legumin-type (brassicas, oat, rice)	[4,7]
Prolamins	Alcohol/water mixtures (e.g., 70% ethanol)	Intra-molecular disulfide bonds, High proportion of proline and glutamine, repetitive motifs in central domains	Gliadins, Zein, Hordein, Secalin	[4,7,8,9,10]
Glutelins	Alkaline solutions	Inter-molecular disulfide bonds, High proportion of proline and glutamine, repetitive motifs in central domains	Glutenins in wheat	[4,7,8,9,10,11]

**Table 2 molecules-25-00873-t002:** Examples of plant proteins with their unique functions in plants, and in food and non-food applications

Utilization	Protein Type (Examples)	Occurrence (Examples)	Function (Examples)	References (Examples)
Living plants	Storage proteins	Seeds, tubers	Growth and nutrition to seedlings and plantlets	[16,17]
	RuBisCo	Chloroplast	Photosynthesis	
	Plasma membrane proteins e.g., surface proteins, globular proteins	Cell membrane, Protein channels	Transport, structural support, ion regulation	[16,18]
Food products	Vicilin	Pea	Heat-induced gelation emulsifying properties	[19,20,21,22]
Bio-based materials	Glutenins	Wheat	Cohesive matrix, gas barrier, strength	[10,23,24,25,26,27,28,29,30,31,32,33,34]
	Gliadins	Wheat	Cohesive matrix, gas barrier, flexibility	

**Table 3 molecules-25-00873-t003:** Methods used to determine protein structures and structure-function relationships.

Method Type	Method (Examples)	Used for/Applications (Examples)	References (Examples)
Microscopy	Transmission Electron Microscopy (TEM)	Three-dimensional (3D) structure of proteins from 2D particle images	[39,48]
	Cryo Electron Microscopy (Cryo-EM)	3D structure of biomacro-molecules in native state	[39,49]
	Tomography	High-resolution 3D images	[50]
	Imaging	Images of individual proteins by low-electron holography	[50,51]
Scattering/ Diffraction, Spectroscopy	Nuclear magnetic resonance Spectroscopy (NMR)	Chemical shifts reflecting conformations of proteins	[52]
	Small-angle X-ray scattering	Shape, conformation and assembly of proteins	[53]
	Wide-angel X-ray scattering	Characterization of structural models, similarities, and changes in atomic packing	[54]
	Fourier Transform Infrared Spectroscopy (FT-IR)	Protein conformation through peak fitting of amide bands	[39,55]
	X-ray crystallography	Atomic resolution of 3D protein structures	[56]
Modeling-Simulation	Template-based	Modeling based on homology	[34]
	Ab initio	Modeling based only on sequence information	[40,41]
	Monte Carlo (MC)	Statistical method evaluating moves of a protein	[42,43]
	Molecular Dynamics (MD)	Solving Newton’s equation of motion	[43]
	Machine aearning (AlphaFold)	Artificial neural network	[45,46,47]

**Table 4 molecules-25-00873-t004:** Structural characteristics of some modeled seed storage proteins.

Protein Type		Structure Prediction Method	Experimental Form	Structure Characteristics	References (Examples)
Globulin	Canavalin	X-ray diffraction	Native	Compact crystal structure with salt bridges and hydrophobic clusters	[93,94,95,96]
	Vicilin	Homology modeling	Amino acid sequence	Trimer	[97]
	Cruciferin	Homology modeling	Amino acid sequence	Hexamer via inter-protomer (IE) disulfide bonds between two trimers	[98]
Prolamin	Gliadins	Dynamic light scattering, cryo-transmission electron microscopy, small-angle X-ray scattering, MD simulations	Gliadin solution at pH 3.0	Dimers of 5.72 nm, aggregated clusters of 30 nm, oligomers of 68 and 103 nm	[103,105]
Glutelin	Glutenins	Small-angle-X-ray scattering	Native/extracted from wheat seed	Intrinsically disordered structure	[34,104]

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
