# Peer review of "Modeling to Understand Plant Protein Structure-Function Relationships—Implications for Seed Storage Proteins"

_molecules, 2020, doi:10.3390/molecules25040873_

Round 1

Reviewer 1 Report

The review from Rasheed et coworkers, "Modeling to understand plant protein structure function relationship implication for grain storage proteins",focuses on various proteins present in plants, with a special emphasis on plant storage proteins and the possibility of using modelling tools to predict proteins structure in order to gain informations on proteins function.

The review as it is written is not well organized and very often misleading. Globally the authors should emphasize better that this is a review about plant storage proteins  and reorganize the structure of the review.

As an example, the authors start with a very long introduction about the classification of plant proteins without clearly mentioning that it is a classification of plant storage proteins. Then they give a description of possible application of plant proteins for food and industrial application again they should  better emphasize, already from the title of the paragraph, that they  are discussing storage proteins.

I would remove the paragraph on Modelling, a too generic paragraph on methods for protein structure determination is of no help for the readers.

It is not clear to me the need of the paragraph on plant proteins modeling, the authors make some examples (allergens, sweet proteins, rubisco) why do they choose these proteins and not other important classes of plant proteins?

Instead I suggest a more thorough literature search to describe the existing literature on structure characterization of storage plant proteins (i.e. literature on globulins) and a subsequently better description of the known structural aspects on these proteins that might be useful for functional studies

Author Response

We are thankful for the valuable comments and scientific critic of the reviewers towards the improvement of the quality of our manuscript. The manuscript has now been thoroughly revised and we have made every attempt to improve the manuscript following every point of the reviewers suggestions. Changes in the manuscript were made according to the editor and reviewers’ comments and the reviewers questions are addressed point by point as below:

The review from Rasheed et coworkers, "Modeling to understand plant protein structure function relationship implication for grain storage proteins",focuses on various proteins present in plants, with a special emphasis on plant storage proteins and the possibility of using modelling tools to predict proteins structure in order to gain informations on proteins function.

The review as it is written is not well organized and very often misleading. Globally the authors should emphasize better that this is a review about plant storage proteins  and reorganize the structure of the review.

Thanks for comments. The review has now been reorganized and restructured in order to more clearly emphasis the message of the paper, what the different sections contain and why they are included in the review. Also, we have clarified which parts and statements are related to the plant storage proteins and emphasis related to the plant storage proteins have been included whenever found suitable to make the review more clear.

As an example, the authors start with a very long introduction about the classification of plant proteins without clearly mentioning that it is a classification of plant storage proteins. Then they give a description of possible application of plant proteins for food and industrial application again they should  better emphasize, already from the title of the paragraph, that they  are discussing storage proteins.

Thanks for the exemplification. Now we have modified the first two paragraphs to make it more clear what statements are related to plant storage proteins and what statements are related to other plant proteins. We have included subheadings to differentiate the topics we are discussing. Furthermore, we have also in each of the subsections related the findings to their usefulness for the seed storage proteins.

I would remove the paragraph on Modelling, a too generic paragraph on methods for protein structure determination is of no help for the readers.

As the other reviewer wanted an increase in this paragraph, we have tried to fulfil both reviewers wishes through delete more generic parts of the paragraph and add some more updates on novel methodologies.

It is not clear to me the need of the paragraph on plant proteins modeling, the authors make some examples (allergens, sweet proteins, rubisco) why do they choose these proteins and not other important classes of plant proteins?

The examples on modeling of plant proteins in this paragraph are those most commonly modelled based on the data bank. So this subchapter is a way to describe what has been done in the modeling of plant proteins area. Such knowledge contribute understanding on opportunities and what has already been done. We have modified the text to make this more understandable.

Instead I suggest a more thorough literature search to describe the existing literature on structure characterization of storage plant proteins (i.e. literature on globulins) and a subsequently better description of the known structural aspects on these proteins that might be useful for functional studies

Literature on the suggested areas has again been searched and added. The paragraph has also been divided into different subsections to better emphasis were knowledge on this topic is today.

With these changes, we hope that the manuscript will now be suitable for publication in Molecules.

Reviewer 2 Report

This paper presents a review of the various applications of plant protein modeling for both structure and function prediction, focusing on plant storage proteins.

I have several concerns regarding this work which must be addressed by the authors before this manuscript can be considered for publication:

At line 19 in the abstract, the authors refer to the "plant protein bank". I assume they meant Protein Data Bank (PDB) as the "plant protein bank" is never referenced in the text, nor could I find it by doing a web search. 

In the beginning of the introduction (lines 33 to 39) the authors use some superlative expressions such as "that we can only dream of", "exceptional functions", "at fantastic rates". Please refrain from using such colloquial terminology in a scientific paper.

The authors also state that spider silk is stronger than steel. For your information it is not: https://phys.org/news/2013-06-spider-silk-nature-stronger-steel.html

At line 38, "made-up" means fictional, not real. I assume the authors meant "man made" or something similar.

At line 42, authors mention "emerging plants" but I do not understand what the term refers to. Do they mean growing/sprouting/germinating plants or "emergent" plants?

At line 75, reference 9 is given in a different style. This happens with other references as well within the paper. Please use the same reference style consistently throughout the paper according to the journal's requirements.

In lines 76-77, the phrasing "being the major storage protein in the majority of legumes" a part from being atrocious also puzzles me: aren't peas and soy bean legumes? The authors should rephrase this whole sentence in a more clear way.

At line 96, "functional and storage functions are key functions" ... please rephrase it.

At lines 101-102, "One type of storage source in these plant organs is built up of proteins, so called storage proteins (Table 2)." This sentence makes no sense to me, please rephrase it in a more comprehensible way.

At lines 104-105, "Additionally, the plants have an organelle not present in other living organisms, the chloroplast, taking care of the photosynthesis (Table 2)." Eukaryotic algahe are not plants but do have chloroplasts!

At lines 147-149, the authors state that "For such proteins, not having any relationship with already posted proteins, ab initio protein structure prediction, i.e., modeling from sequence information alone, is an option." It is the only option, not an option! Unless the authors mean that experimental techniques could be used. In any case, the sentence should be rephrased in a more intelligible way.

I do not agree with the classification of the in silico protein modeling methods described by the authors. Monte Carlo and Molecular Dynamics simulations are both types of ab initio protein structure prediction. Moreover, in lines 150-152, the authors state that the main drawback of ab initio modeling methods is that their accuracy is low and that their applicability is limited to small proteins with up to 100 residues. They are completely ignoring the advances in the computational structure prediction field of the past decade (the reference 39 they provide is a 2009 paper). They also fail to mention any machine learning (or deep learning) based prediction method.  I suggest the authors to have a look at the work of Mohammed AlQuraishi, namely AlphaFold. The 3D models of proteins that AlphaFold generates are far more accurate than those generated with any previous method.

Table 3 does not enlist template-based (homology) modeling in the modeling-simulation. At this point, I advise the authors to completely restructure section 3. Now, I understand that there might not be many instances of plant protein structures that have been modeled with the latest methods, but this is no reason for completely ignoring the past 10 years of advancements in the protein structure prediction field.

At lines 229-230, "storage proteins are the major protein types in the majority of all plants with storage organs", please rephrase!

Please add the separating horizontal lines in tables 1 and 4, as you did in tables 2 and 3.

Even after reading the paper several times, it is still unclear to me what is the main focus of this work. What are the critical issues? What are the most probable future directions? The last paragraph of section 5 (lines 278-299) seems to briefly address these issues, but it should be significantly expanded. I suggest the authors to provide some insights on what the results of applying the latest protein structure prediction methods would be. 

Finally, I suggest the authors to seek help from someone with good English language proficiency for proofreading their paper.

Author Response

We are thankful for the valuable comments and scientific critic of the reviewers towards the improvement of the quality of our manuscript. The manuscript has now been thoroughly revised and we have made every attempt to improve the manuscript following every point of the reviewers suggestions. Changes in the manuscript were made according to the editor and reviewers’ comments and the reviewers questions are addressed point by point as below:

This paper presents a review of the various applications of plant protein modeling for both structure and function prediction, focusing on plant storage proteins.

I have several concerns regarding this work which must be addressed by the authors before this manuscript can be considered for publication:

At line 19 in the abstract, the authors refer to the "plant protein bank". I assume they meant Protein Data Bank (PDB) as the "plant protein bank" is never referenced in the text, nor could I find it by doing a web search. 

We thank the reviewer for being attentive. Changes have been done accordingly.

In the beginning of the introduction (lines 33 to 39) the authors use some superlative expressions such as "that we can only dream of", "exceptional functions", "at fantastic rates". Please refrain from using such colloquial terminology in a scientific paper.

Superlative expressions have been changed.

The authors also state that spider silk is stronger than steel. For your information it is not: https://phys.org/news/2013-06-spider-silk-nature-stronger-steel.html

Changes have been made accordingly

At line 38, "made-up" means fictional, not real. I assume the authors meant "man made" or something similar.

Changes made accordingly.

At line 42, authors mention "emerging plants" but I do not understand what the term refers to. Do they mean growing/sprouting/germinating plants or "emergent" plants?

Changes done.

At line 75, reference 9 is given in a different style. This happens with other references as well within the paper. Please use the same reference style consistently throughout the paper according to the journal's requirements.

Thanks for comment. References have been checked and changed.

In lines 76-77, the phrasing "being the major storage protein in the majority of legumes" a part from being atrocious also puzzles me: aren't peas and soy bean legumes? The authors should rephrase this whole sentence in a more clear way.

Sentence have been rephrased.

At line 96, "functional and storage functions are key functions" ... please rephrase it.

Sentence have been rephrased.

At lines 101-102, "One type of storage source in these plant organs is built up of proteins, so called storage proteins (Table 2)." This sentence makes no sense to me, please rephrase it in a more comprehensible way.

Sentence have been rephrased.

At lines 104-105, "Additionally, the plants have an organelle not present in other living organisms, the chloroplast, taking care of the photosynthesis (Table 2)." Eukaryotic algahe are not plants but do have chloroplasts!

Thanks for comments. Changes have been made.

At lines 147-149, the authors state that "For such proteins, not having any relationship with already posted proteins, ab initio protein structure prediction, i.e., modeling from sequence information alone, is an option." It is the only option, not an option! Unless the authors mean that experimental techniques could be used. In any case, the sentence should be rephrased in a more intelligible way.

Sentence rephrased.

I do not agree with the classification of the in silico protein modeling methods described by the authors. Monte Carlo and Molecular Dynamics simulations are both types of ab initio protein structure prediction. Moreover, in lines 150-152, the authors state that the main drawback of ab initio modeling methods is that their accuracy is low and that their applicability is limited to small proteins with up to 100 residues. They are completely ignoring the advances in the computational structure prediction field of the past decade (the reference 39 they provide is a 2009 paper). They also fail to mention any machine learning (or deep learning) based prediction method.  I suggest the authors to have a look at the work of Mohammed AlQuraishi, namely AlphaFold. The 3D models of proteins that AlphaFold generates are far more accurate than those generated with any previous method.

Thanks for useful comments which have been taken into consideration and changes made. The other reviewer wanted the whole section 3 to be deleted, so we tried to decrease on the info in this section, simultaneously as increasing with novel methods.

Table 3 does not enlist template-based (homology) modeling in the modeling-simulation. At this point, I advise the authors to completely restructure section 3. Now, I understand that there might not be many instances of plant protein structures that have been modeled with the latest methods, but this is no reason for completely ignoring the past 10 years of advancements in the protein structure prediction field.

Template based and machine learning included in Table 3. And info on usefulness for seed storage proteins included in text.

At lines 229-230, "storage proteins are the major protein types in the majority of all plants with storage organs", please rephrase!

Sentence rephrased.

Please add the separating horizontal lines in tables 1 and 4, as you did in tables 2 and 3.

Horizontal lines added.

Even after reading the paper several times, it is still unclear to me what is the main focus of this work. What are the critical issues? What are the most probable future directions? The last paragraph of section 5 (lines 278-299) seems to briefly address these issues, but it should be significantly expanded. I suggest the authors to provide some insights on what the results of applying the latest protein structure prediction methods would be. 

Thanks for comments. The paper has now been thoroughly gone over to increase the understanding of the paper and to clarify purpose of the paper and possible uses of modeling for plant proteins and more specifically for seed storage proteins.

Finally, I suggest the authors to seek help from someone with good English language proficiency for proofreading their paper.

A native English speaking person has now checked the language in the paper.

With these changes, we hope that the manuscript will now be suitable for publication in Molecules.

Round 2

Reviewer 1 Report

In the resubmitted version of the manuscript "Modeling to understand plant protein structure-function relationships – implications for grain storage proteins" the authors have tried to fullfill the requirements  of the reviewers but in my opinion the resubmitted manuscript is not very much improved.

All the general parts about "plant proteins" weaken the paper, a mix of general informations not exaustive that do not bring any significant contribution to the field. I would go straight for a review only about storage proteins highlighting in more details  functional and structural aspects.

In the paragraph on the modelling of plant proteins,  when the authors talk about  "Enzymes", they mix up modelling of metabolic pathways and molecular modelling two aspects conceptually very different. In the paragraph on sweet proteins as experimental techniques used to calculate structures, beside XRAY, also NMR has been very extensively used.

In the text the authors talk of 2000 plant proteins structures deposited in PDB but they are more than 5000.

In the general description of modelling methods the description of homology modelling the authors talk of aminoacid homology but it is correct to say sequence homology

It is a bit extreme to say that  "Monte Carlo (MC) and Molecular Dynamics can be seen as an ab initio type of protein modeling"

The paragraph about the modelling on plant proteins should be more detailed on existing literature and structural aspects described more exstensively. It is not enough to have a figure at the end of the conclusions not even discussed in detail.

Moreover  the authors should really go carefully through the text because some parts of the manuscript are very sloppy.

Author Response

We are thankful for the valuable comments and scientific critic of the reviewers, which we have tried to use constructively to improve the quality of our manuscript. The manuscript has now been thoroughly revised and we have made every attempt to improve the manuscript following every point of the reviewers suggestions. Changes in the manuscript were made according to the editor and reviewers’ comments and the reviewers questions are addressed point by point as below:

In the resubmitted version of the manuscript "Modeling to understand plant protein structure-function relationships – implications for grain storage proteins" the authors have tried to fullfill the requirements  of the reviewers but in my opinion the resubmitted manuscript is not very much improved.

Our intention was and also in this round is to try to use the reviewers comments constructively to revise and improve the manuscript. We have thoroughly gone through comments from reviewers and further improved the manuscript. With changes made in this round, we think that the reviewer will find the manuscript enough improved for publication.

All the general parts about "plant proteins" weaken the paper, a mix of general informations not exaustive that do not bring any significant contribution to the field. I would go straight for a review only about storage proteins highlighting in more details  functional and structural aspects.

Our intention here, while planning the manuscript was to compare what has been done in the field for other characters and what is the situation for plant storage proteins. The work carried out on plant storage proteins is very limited and to focus on only that part will make the paper rather thin. Also, there is a point in comparing and learn from other characters, to our opinion.

In the reviewing process, it seems like the two reviewers have quite contradictory opinions on what to do with the manuscript, where reviewer 2 wanted the part 3 about modeling and the more general parts on plant proteins to be increased and improved. Thus, this contradictory opinions make it also very difficult to follow reviewer 1 suggestion to take away these parts and focus on only the storage proteins.

To try to take advantage of both reviewers comments and suggestions, and to improve the manuscript after our best intentions, we have, in the revised version, therefore kept the more general parts and the part 3, but have tried to decrease focus on these and to make it more clear what are general and what are specific for storage proteins. We have also tried to include more information of what can we learn from general parts that can be of relevance for storage proteins. We hope that our attempts to follow both reviewers good advices and our intention to improve the manuscript, really has made the manuscript more clear and better so that it suits for publication in Molecules.

In the paragraph on the modelling of plant proteins,  when the authors talk about  "Enzymes", they mix up modelling of metabolic pathways and molecular modelling two aspects conceptually very different. In the paragraph on sweet proteins as experimental techniques used to calculate structures, beside XRAY, also NMR has been very extensively used.

Thanks for useful comments. These parts have been made more clear in the present revised manuscript.

In the text the authors talk of 2000 plant proteins structures deposited in PDB but they are more than 5000.

As these is a data base continuously updated, we have made a new search and corrected the numbers to more recent ones.

In the general description of modelling methods the description of homology modelling the authors talk of aminoacid homology but it is correct to say sequence homology

Corrected accordingly.

It is a bit extreme to say that  "Monte Carlo (MC) and Molecular Dynamics can be seen as an ab initio type of protein modeling"

We agree, although this was a requisite from the other reviewer. Text has been revised to make the statement “milder”.

The paragraph about the modelling on plant proteins should be more detailed on existing literature and structural aspects described more exstensively. It is not enough to have a figure at the end of the conclusions not even discussed in detail.

Changes made accordingly to improve this part.

Moreover  the authors should really go carefully through the text because some parts of the manuscript are very sloppy.

Text has been thoroughly gone over and also language edited by a professional language editor to improve the text not being sloppy.

Reviewer 2 Report

I have checked the revised version of the manuscript and I am satisfied with the improvements the authors made. In my opinion, the article is now ready for publication.

Author Response

Thanks for positive comments.